# Diagnosis of Delayed Post-Hypoxic Leukoencephalopathy (Grinker’s Myelinopathy) with MRI Using Divided Subtracted Inversion Recovery (dSIR) Sequences: Time for Reappraisal of the Syndrome?

**DOI:** 10.3390/diagnostics14040418

**Published:** 2024-02-14

**Authors:** Gil Newburn, Paul Condron, Eryn E. Kwon, Joshua P. McGeown, Tracy R. Melzer, Mark Bydder, Mark Griffin, Miriam Scadeng, Leigh Potter, Samantha J. Holdsworth, Daniel M. Cornfeld, Graeme M. Bydder

**Affiliations:** 1Mātai Medical Research Institute, Tairāwhiti Gisborne 4010, New Zealand; gil@neuropsychiatrygilnewburn.co.nz (G.N.); m.scadeng@auckland.ac.nz (M.S.); s.holdsworth@matai.org.nz (S.J.H.);; 2Department of Anatomy and Medical Imaging, Faculty of Medical and Health Sciences & Centre for Brain Research, University of Auckland, Auckland 1010, New Zealand; 3Department of Medicine, University of Otago, Christchurch 8011, New Zealand; 4New Zealand Brain Research Institute, Christchurch 8011, New Zealand; 5Insight Research Services Associated, Gold Coast 4215, Australia; 6Department of Radiology, University of California San Diego, San Diego, CA 92093, USA

**Keywords:** delayed post-hypoxic leukoencephalopathy, Grinker’s myelinopathy, targeted MRI of the brain, divided subtracted inversion recovery sequences, T_1_-bipolar filter, white matter disease of the brain

## Abstract

**Background:** Delayed Post-Hypoxic Leukoencephalopathy (DPHL), or Grinker’s myelinopathy, is a syndrome in which extensive changes are seen in the white matter of the cerebral hemispheres with MRI weeks or months after a hypoxic episode. T_2_-weighted spin echo (T_2_-wSE) and/or T_2_-Fluid Attenuated Inversion Recovery (T_2_-FLAIR) images classically show diffuse hyperintensities in white matter which are thought to be near pathognomonic of the condition. The clinical features include Parkinsonism and akinetic mutism. DPHL is generally regarded as a rare condition. **Methods and Results:** Two cases of DPHL imaged with MRI nine months and two years after probable hypoxic episodes are described. No abnormalities were seen on the T_2_-FLAIR images with MRI, but very extensive changes were seen in the white matter of the cerebral and cerebellar hemisphere on divided Subtraction Inversion Recovery (dSIR) images. dSIR sequences may produce ten times the contrast of conventional inversion recovery (IR) sequences from small changes in T_1_. The clinical findings in both cases were of cognitive impairment without Parkinsonism or akinetic mutism. **Conclusion:** The classic features of DPHL may only represent the severe end of a spectrum of diseases in white matter following global hypoxic injury to the brain. The condition may be much more common than is generally thought but may not be recognized using conventional clinical and MRI criteria for diagnosis. Reappraisal of the syndrome of DPHL to include clinically less severe cases and to encompass recent advances in MRI is advocated.

## 1. Introduction

Acute sequelae seen in the brain with MRI after global hypoxia due to severe drug overdoses, asphyxiation and other causes include territorial and border zone infarctions, as well as damage to grey matter [1]. Chronic changes seen with MRI in the brain after global hypoxia usually follow from the acute changes and typically include later stages of infarction and grey matter changes, but rarely, Delayed Post-Hypoxic Leukoencephalopathy (DPHL) may develop. This condition selectively involves white matter [2,3,4,5,6]. Its clinical features include an initial recovery period followed by deterioration typically beginning 2–4 weeks later. Classically, the presentation at this stage follows two common patterns, (i) Parkinsonism and (ii) akinetic muteness, although there may be other features.

DPHL is sufficiently rare for individual case reports and small series of 2–5 patients to be published [4,5,6,7,8,9,10,11,12,13]. The term Grinker’s myelinopathy is used to describe the condition, which is named after Roy Richard Grinker Sr (1900–1993), who described it in 1925/26 when he was an instructor in neurology at Northwestern University in Evanston, Illinois [14].

The MRI diagnosis of DPHL is usually made with T_2_-weighted Fast Spin Echo (T_2_-wFSE) and/or T_2_-Fluid Attenuated Inversion Recovery (T_2_-FLAIR) sequences, which show diffuse hyperintensities in the central white matter of the cerebral hemispheres. U-fibres are spared, as is cerebellar white matter [3,4,5,6]. These MRI features are regarded as “near pathognomonic” for DPHL [4].

We report two cases of suicide attempts probably associated with significant hypoxia in patients who were examined with an MRI nine months and two years, respectively, after their suicide attempts. No abnormalities were seen on their T_2_-FLAIR images, but extensive changes were seen in the white matter of their cerebral and cerebellar hemispheres using divided Subtracted Inversion Recovery (dSIR) sequences. These sequences are sensitive to changes in T_1_ and may show ten or more times the contrast seen with conventional inversion recovery (IR) sequences when imaging small increases in T_1_ from normal levels due to disease [15].

We raise the possibility that widespread white matter disease of the brain may be a relatively common late complication of global hypoxia but may not be recognized with conventional MRI sequences. Classical clinical descriptions of DPHL may only apply to more severe forms of the disease spectrum that produce changes seen with conventional MR sequences. The syndrome of DPHL may need reappraisal to accommodate findings with more sensitive MRI sequences and include neuropsychiatric symptoms and signs that are less severe than Parkinsonism and akinetic mutism.

## 2. Theory

In this section, the theory underlying the use of dSIR sequences is described in condensed form. The description is included to make this paper self-contained and to provide a basis for the interpretation of dSIR images as well as the discussion of their features. More detailed descriptions of dSIR sequences and images are included in previous papers [15,16].

The mechanism underlying dSIR sequence contrast is shown in Figure 1 and Figure 2.

Two magnitude reconstructed IR T_1_ filters with different Tis, TIshort = TI_s_ and TIintermediate = TI_i_, are shown in Figure 1A. T_1_ filters are plots of signals against T_1_ for MR sequences. When the second IR T_1_ filter is subtracted from the first IR T_1_ filter, this produces the Subtracted IR (SIR) T_1_ filter in Figure 1B. In the central region or middle Domain (mD) of the SIR T_1_ filter between the two vertical dashed lines in Figure 1B, the slope of the filter is about double that of the IR T_1_ filters shown in Figure 1A.

The two T_1_ filters shown in Figure 1A can be added to give the Added IR (AIR) T_1_ filter, as shown in Figure 1C. In its mD, which is bounded by the vertical dashed lines, the signal is reduced to about 0.20 of its value of 1 at T_1_ = 0.

Figure 2A shows a dSIR T_1_-bipolar filter in which the SIR T_1_ filter in Figure 1B is divided by the AIR T_1_ filter in Figure 1C. The resulting dSIR T_1_-bipolar filter shows a highly positive nearly linear slope in its mD, where its slope is about ten times that of the IR T_1_ filters shown in Figure 1A.

Figure 2B compares the contrast produced by the S_TIs_ IR T_1_ filter (pink) to that from the SIR T_1_ filter (blue) from the same increase in T_1_ (∆T_1_) produced by disease (horizontal green arrow, ∆T_1_). Using the small change approximation of differential calculus, ∆T_1_ is multiplied by the slopes of the respective T_1_ filters (red lines) to produce the differences in signal ∆S or contrasts produced by the S_TIs_ and SIR T_1_ filters shown by the vertical pink and blue arrows on the right. The SIR T_1_ filter generates about twice the contrast (blue arrow) of the S_TIs_ dIR T_1_ filter (pink arrow) from the same small increase in T_1_.

Figure 2C compares the contrast produced by the S_TIs_ IR T_1_ filter (pink) to that produced by the dSIR T_1_-bipolar filter (blue). The increase in T_1_ (∆T_1_) produced by disease (horizontal green arrow, ∆T_1_) is multiplied by the slopes of the respective S_TIs_ IR and dSIR T_1_ filters (red lines) to produce the differences in signal ∆S or contrasts generated by the two filters. This is shown by the vertical pink and blue arrows on the right. For the same increase in T_1_, ∆T_1_, the dSIR T_1_-bipolar filter, produces about ten times more contrasts than the S_TIs_ IR T_1_ filter, such as Magnetization-Prepared Rapid Acquisition Gradient Echo (MP-RAGE).

To produce the large increase in contrast shown in Figure 2C, dSIR sequences need to be correctly targeted at the normal tissue of interest and the small increases in the tissue T_1_ produced by disease.

## 3. Methods

With the approval of the Auckland Hospital Research Ethics Committee (approval number AHRECAH1006, 2021) and with informed consent from each subject, MRI scans were performed on an 18-year-old normal male volunteer, a 19-year-old male patient nine months after a severe drug overdose and a 20-year-old male patient two years after attempted suicide by hanging. A 3T scanner (General Electric Healthcare, Chicago, IL, USA) was used. Two-dimensional IR FSE sequences were performed with a TI_s_, chosen to null the shortest T_1_ of normal white matter, and a longer TI_i_, chosen to produce narrow mD dSIR images targeted at small increases in T_1_ in white matter from normal levels, as illustrated in Figure 2C. Two-dimensional and 3D T_2_-FLAIR images were also acquired, as described in Table 1. With 3D acquisitions, adjacent T_2_-FLAIR images were added to give similar slice thicknesses to those of the dSIR images to assist a comparison of the two types of images.

## 4. Results


**Normal Volunteer**


MRI findings:

Figure 3 shows narrow mD dSIR images from the normal 18-year-old volunteer. His white matter is normal and shows a low signal (dark) appearance with a mid-grey appearance in and around the corticospinal tracts. Normal high signal boundaries (white lines) are seen at junctions between white and grey matter.


**Case 1**


Case history:

Case 1 was a 19-year-old male patient who was the product of a pregnancy with a prolonged second stage of labour, and he required some resuscitation. His development proceeded within normal limits for physical activity, but there were social and cognition issues consistent with high functioning autism spectrum disorder during early adolescence. In his penultimate year of high school, there were attentional difficulties in an otherwise intelligent young man. At age 19 years, he took a life-threatening drug overdose. He consumed unknown, but large, amounts of amitriptyline, paracetamol, codeine phosphate, zopiclone, clonazepam and promethazine. The overdose was probably taken late in the evening, but he was not found until around 2:00 p.m. the following day when he was deeply unconscious. There were pressure sores on the left side of his body, and he suffered compression injuries to the neural structures of his left arm. He probably survived because there had been clumping of drugs in his stomach with a slow release and absorption of them. After an emptying of his stomach contents, he slowly regained consciousness and improved functionally. Nine months later, he retained his prior intellect but had some slowing of information processing as well as reductions in his attentional capacity and overall function.

MRI findings:

Figure 4 and Figure 5 show T_2_-FLAIR images (upper rows) with positionally matched dSIR images (lower rows). No abnormality is seen in white matter on the T_2_-FLAIR images, but very extensive high signal abnormalities are seen in white matter on the corresponding dSIR images. In Figure 4, there is sparing of white matter in the anterior central corpus callosum and adjacent forceps minor, which have a dark appearance (long white arrows). There is also some sparing of the posterior central corpus callosum. In Figure 5, there is sparing of the peripheral white matter of the cerebral hemispheres, which have a dark appearance (long white arrows). Small focal lesions of a relatively increased signal are also seen in the white matter in Figure 5 (grey arrows) on the dSIR images. High signal boundaries between the white and grey matter are seen, but these are less obvious in many areas because of the high signal present in much of the white matter.

Overall, the abnormalities are bilateral and symmetrical and involve most of the white matter of both the cerebral and cerebellar hemispheres. The appearances in the patient are strikingly different from those in the normal control shown in Figure 3, where normal white matter has a low signal (dark appearance), apart from the corticospinal tracts and areas adjacent to them, which are mid-grey.


**Case 2**


Case history:

Case 2 was a male aged 20 years at the time of MRI scanning. As far as is known, he was the product of a normal pregnancy and delivery, with no alcohol or illicit drug consumption by his mother, although she did have a subsequent history of significant methamphetamine abuse. Cultural adoption took place at seven months with a supportive formative environment. Although it was never formally diagnosed, his history strongly suggested the presence of attention deficit hyperactivity disorder of combined type.

At 18 years of age, following a relationship break-up, the patient began consuming large amounts of alcohol in a binge pattern and using cannabis. In what was probably an intoxicated state, he made a suicide attempt by hanging. He lost anal sphincter tone during this event. He was found by chance and was transported to hospital by ambulance where he was admitted to the Intensive Care Unit, although he has no subsequent recollection of this. On subsequent examination, he had a period of at least several hours of retrograde amnesia and an indeterminate period of anterograde amnesia. He was permitted to discharge himself from hospital the following day, presumably still in a state of delirium, given his lack of memory.

Following this event, he slowly redeveloped an awareness of himself over the next few days and became aware that his information processing had slowed. He reached cognitive overload much more readily. Attentional function had deteriorated further, and he struggled to divide and alternate attention or cope with any environment where the stimulus load increased beyond what was essentially a one-to-one interaction. Even then, if material was presented at more than one item per statement, he was unable to follow it, although he processed single items adequately. Shifting mental sets became difficult for him with significant fight/flight reactivity, if this was demanded of him without prior warning. His higher-order cognitive function fell away, with a fall-off in pragmatic communication skills. He struggled with metacognition in the moment, although his post-hoc awareness seemed adequate. He also described subtle second-order theory-of-mind difficulties. His sleep–awake cycle became disrupted, although it was difficult to discern the exact pattern of this (he had always struggled to settle at night as part of his attention deficit disorder).

After the suicide attempt, he remained in a state of distress, both due to the loss of a relationship and increasingly due to his struggles to cope cognitively and socially. He continued to abuse alcohol and cannabis in a binge manner (it was for the management of this that he presented for assessment). At no stage over the two years following his suicide attempt had any health professional appeared cognizant of his premorbid attention deficit disorder or the possibility that he had suffered post-hypoxic brain damage as a result of his attempted suicide.

MRI findings:

Figure 6 and Figure 7 show T_2_-FLAIR images (upper rows) with positionally matched dSIR images (lower rows). No abnormalities are seen in the white matter on the T_2_-FLAIR images, but very extensive high signal abnormalities are seen in the white matter of the corresponding dSIR images. There is relative sparing of the anterior central corpus callosum and, to a lesser extent, the posterior central corpus callosum. There is also some sparing of the peripheral white matter in the cerebral hemispheres.

Overall, the MRI findings in Case 2 are very similar to those in Case 1, as shown in Figure 4 and Figure 5. They have been described as a whiteout sign. This often involves 80% or more of the white matter in the cerebral and cerebellar hemispheres having an abnormal high signal appearance.

## 5. Discussion

### 5.1. Clinical and MRI Summary

One patient with a history of a severe drug overdose and another with a history of attempted suicide by asphyxia were examined with MRI nine months and two years afterwards using T_2_-FLAIR and dSIR sequences. In both cases, no abnormalities were seen in their white matter with T_2_-FLAIR sequences, but very extensive changes were seen in their white matter using dSIR sequences. In both cases, the changes were bilateral and symmetrical and involved the cerebral and cerebellar hemispheres. There was sparing or relative sparing of the anterior central corpus callosum and adjacent white matter and, to a lesser extent, the posterior central corpus callosum. There was also sparing of the white matter in the periphery of the cerebral hemispheres.

### 5.2. Grinker’s Myelinopathy

The syndrome of myelinopathy after hypoxia was first described by Roy Richard Grinker Sr. when he was 25/26 years old, and it bears his name. This condition has also been described as DPHL, delayed hypoxic leukoencephalopathy [5], delayed postanoxic encephalopathy [6] and delayed post-hypoxic encephalopathy [7]. It is regarded as a rare disease, and case reports and small series of a few patients (e.g., typically 2–5, with a maximum in one series of 12 patients) are all that is available in the literature. Roy Richard Grinker Sr. (1900–1993) was an American neurologist and psychiatrist who practised in Chicago [14,17,18]. He published his first major book, “Grinker’s Neurology”, in 1933. It went through seven editions. He was psychoanalysed by Sigmund Freud in Vienna in 1935–1936 and followed a successful career in psychiatry. He was particularly interested in combat fatigue during and after World War II and co-wrote a book about this [19]. His father was also a psychiatrist, as was his son Roy Richard Grinker Jr. (1927–2022). His grandson, Roy Richard Grinker (1961-), an anthropologist, has published on autism and how culture created the stigma of mental illness [20].

### 5.3. Clinical Features of DPHL

The classical features of DPHL are a global hypoxic event, recovery (typically from coma) after the hypoxic event and deterioration with two typical clinical presentations at this stage, namely, (i) Parkinsonism and (ii) akinetic mutism [3]. There may also be dystonic posturing, agitation, hallucinations and unusual behaviour. Patients may exhibit slow verbal responses with impaired cognition and emotional lability. Akinetic mute patients may be profoundly apathetic and may display functional bowel and bladder incontinence, minimal primitive responses to pain and pathological laughter or crying. Physical examinations may include frontal release and corticospinal tract signs. Early-on cognitive symptoms may be so profound that detailed testing is not possible. The CSF myelin basic protein is usually elevated as a result of active demyelination.

The two cases described in this paper did not have the neurological features of Parkinsonism or akinetic mutism typical of the classical syndrome. They showed cognitive impairment. In addition, the patients were much younger (18 and 20 years) than those described in the literature, who were generally over 30 and with average ages over 60 in some series.

### 5.4. CT and MRI Features of DPHL

Historically, the diagnosis of DPHL was initially made without imaging. Modern imaging of the brain has gone through distinct periods, and the diagnosis of DPHL has evolved with this. These stages have included (i) CT, (ii) MRI and (iii) advanced MRI, as used in this paper.

(i).CT: This typically showed reductions in X-ray attenuation in white matter in DPHL, as described by Lee and Marsden and incorporated, by them, into disease descriptions [3].(ii).MRI: Findings with conventional MRI in DPHL from 1991–2015 were reviewed by Zamora et al. [7], together with five of the authors’ own cases. There were 22 reports on a total of 41 patients. The largest series was 12 cases of carbon monoxide poisoning [21]. The number of cases in other reports was 1–5. Overall, there were a total of 24 cases of carbon monoxide poisoning and 19 cases of drug overdoses, almost all of which were with opioids or benzodiazepines. The most significant and diagnostic MRI finding was extensive and confluent hyperintensities on T_2_-wSE and/or T_2_-FLAIR images. These were predominant in the periventricular white matter and the centrum semiovale and were bilateral and symmetrical. In all patients, the hyperintensities spared the U-fibres, and there was no involvement of the cerebellum in the authors’ series of five cases. The studies in the literature also did not report abnormalities in the cerebellum.

In two of five patients reported by the authors in their own series [7], diffusion restriction matched the extent of the T_2_-FLAIR hyperintensities, while in the other three patients, diffusion abnormalities were less extensive. Diffusion imaging was inconsistently performed in the 22 papers reviewed and was either restricted or normal [7]. Restricted diffusion is thought to arise from the trapping of water in abnormal myelin.

In the two cases described in this paper, T_2_-FLAIR sequences showed no abnormality, but extensive changes in white matter were seen on the dSIR images.

### 5.5. Pathology

Plum et al. reported autopsy findings of severe diffuse demyelination partially sparing subcortical U-fibres. Tissue examples demonstrated the presence of reactive astrocytes and lipid-laden macrophages. Axons and oligodendroglia were present [2].

The syndrome did not include imaging initially but has evolved over its near 100-year history to include modern imaging in the absence of pathology in most cases. The pathological entities detected by imaging may include oedema, demyelination, dysmyelination, neuroinflammation and degeneration, covering a wider spectrum of diseases than in the original description [22,23]. These pathological features may be manifest as generalized processes involving white matter in a bilateral symmetrical manner.

Hypoxia is central to the causation of the disease. There is some evidence that CO may be specifically toxic for myelin and may contribute to more severe diseases.

The biphasic nature of the DPHL raises the question as to whether the disease is immunomodulated and indicates that the delayed white-matter changes are due, at least in part, to the neuroinflammation that is secondary to this.

The pattern of the seen imaging has similarity to genetically determined diseases, such as metachromatic leukodystrophy.

### 5.6. dSIR Sequences

dSIR sequences can provide a ten-times greater amplification of the contrasts of conventional T_1_-weighted IR sequences and may produce contrasts from small increases in T_1_, which accompany small increases in T_2_ but are insufficient to produce diagnostic contrasts with conventional T_2_-wSE and T_2_-FLAIR sequences (Figure 1 and Figure 2).

dSIR sequences also produce high signal boundaries between the white matter and grey matter. These are evident as well defined “etched” lines on images. They are produced where partial volume effects at boundaries between the white and grey matter result in mixed tissue voxels, which have T_1_s precisely corresponding to the maximal signal value shown on the T_1_-bipolar filter (Figure 2).

### 5.7. Targeting

DPHL is an application of targeted MRI par excellence in which normal white matter and small increases in T_1_ are targeted with dSIR sequences to produce high contrasts and show an abnormal whiteout appearance.

In principle, the same approach can be applied to other diseases which affect white matter. It can also, in principle, be applied to other tissues where small changes in T_1_ may be present but are below the threshold necessary to demonstrate useful contrasts with conventional sequences. The changes may be widespread within the tissue and have a significant clinical impact in spite of their small size.

### 5.8. Implementation of dSIR Sequences

The IR Fast Spin Echo (FSE) sequences used to produce dSIR images are available on most clinical MR systems. They can be run with spatial resolutions similar to those of T_2_-wSE and T_2_-FLAIR sequences. The initial TI_s_ is designed to null white matter with the shortest T_1_ of clinical interest, although the chosen TI can be somewhat less than this value. The second TI_i_ is longer and aims to produce an mD which includes small increases in T_1_ in white matter due to disease. The contrast amplification produced by the dSIR sequence increases as the difference in TI, DTI, is decreased. This continues until the sequence becomes noise and/or artefact limited.

The processing of the source IR images to produce dSIR images only involves basic arithmetic and can be implemented in MATLAB.

dSIR imaging can be integrated into workflow patterns with conventional MP-RAGE, T_2_-wSE and/or T_2_-FLAIR imaging. These sequences show effects due to larger changes in T_1_ and/or T_2_, while dSIR sequences show effects due to smaller changes in T_1_, which are beneath the threshold for detection with conventional IR sequences.

### 5.9. Validation of dSIR Imaging Findings

The original clinical description of DPHL was validated pathologically at postmortem. This may have biased the definition towards the severe end of the disease spectrum. As with other diseases of the CNS, imaging has been used in in vivo studies in humans. The validation of the imaging findings is based on the following:(i)Theory, as described using Figure 1 and Figure 2, which provides a coherent account of the contrast seen on dSIR images.(ii)Normal controls, which consistently show low signals in white matter with dSIR sequences.(iii)Boundaries between the white and grey matter: These are a unique and distinctive feature of dSIR images and can be explained using the model in Figure 1 and Figure 2. They are a consistent finding in normal subjects and patients and support the validity of the model.(iv)The presence of whiteout signs, very similar to those shown in Figure 4, Figure 5, Figure 6 and Figure 7, in other diseases such as mTBI [24], methamphetamine dependency [14] and, to a lesser extent, MS [15]. The whiteout sign probably has multiple causes, but a common feature may be neuroinflammation, which may be seen in many different diseases.(v)There is also a consistency between lesions with large changes in the T_2_ (as well as T_1_) seen on T_2_-FLAIR images in the same location on dSIR images. The features include a high signal boundary around the lesion on dSIR images, as predicted by a theory [15].(vi)dSIR sequences have been run on GE, Philips and Siemens MR systems at both 1.5 T and 3.0 T, and images obtained in this way have been consistent for contrasts and high signal white matter–grey matter boundaries. Failure modes for the dSIR sequence have been described and are relatively easy to identify and so ensure consistent results with different machines in both volunteers and patients [15].(vii)There is a correspondence between U-fibre sparing observed pathologically as well as with conventional imaging and the sparing of peripheral white matter observed on dSIR images.

Animal studies with imaging including experimental hypoxia may be necessary to provide histological validation and a detailed correlation of imaging findings with this.

### 5.10. Additional Imaging Options

Myelin-specific sequences, such as DESIRE [25] and STAIR [26], may be very helpful in producing a specific diagnosis of demyelination. The degree to which the findings are reversible will need to be determined. It will be necessary to determine the natural history of the dSIR changes as well as the effect of treatment on them.

There are also technical advances in AI and image noise reduction and registration which may improve the quality of dSIR images.

Additional imaging with dSIR images can be created synthetically from T_1_ maps. It is also possible to synthetically create narrower mD images from wider mD images. The T_1_ maps may be produced by MR fingerprinting and other methods, such as Actual Flip Angle imaging [27], which do not involve an IR sequence so that the more general term T_1_-bipolar filter (T_1_-BLAIR) imaging may be preferred to describe imaging utilizing a T_1_-bipolar filter. The term includes direct acquisitions (such as with dSIR) as well as synthetic imaging utilizing T_1_ maps generated in different ways.

### 5.11. Normal-Appearing White Matter

In this study, normal-appearing white matter (NAWM) on T_2_-FLAIR sequences in Cases 1 and 2 showed obvious abnormalities with dSIR sequences. NAWM was described first pathologically and then in MRI in studies of Multiple Sclerosis (MS), as described in 1989 [28]. The assumption with NAWM in MRI is that there may be an underlying disease present that is not shown with conventional sequences. The usual ways of pursuing these changes in the past has been by the use of different tissue properties such as magnetization transfer and magnetic resonance spectroscopy, but neither of these approaches has become established in routine clinical practice. The present study differs from these approaches by exploiting T_1_, one of the tissue properties used with conventional clinical sequences in the form of the MP-RAGE sequences. Small changes in T_1_, which are insufficient to produce contrasts with MP-RAGE sequences, can be amplified by the dSIR sequence and produce obvious contrasts.

### 5.12. Clinical Importance of DPHL

The findings of the evidence of brain damage with dSIR sequences are important in this group of patients who generally already have diagnoses of psychiatric disorders. After severe drug overdoses or other hypoxic events, their underlying brain damage may not be recognized, particularly when conventional MRI-examination findings are negative.

### 5.13. Reappraisal of the Syndrome of DHPL

The very different clinical and imaging features of the two cases described in this paper to those in classical descriptions of DPHL raise the possibility that these cases may be examples of a different condition rather than a less severe form of DPHL. In favor of the patients described in this paper having DPHL is (i) the common feature of an initiating global hypoxic event, (ii) the delayed onset and persistence of symptoms as well as (iii) the predominant white-matter involvement in bilateral and symmetrical form with similar sparing of U-fibres/peripheral white matter. There are differences in the sparing of the central corpus callosum and the involvement of the cerebellum seen on dSIR images, but these may be additional supplementary features that were not recognized previously. Overall, it appears more likely that the cases described in this paper are less severe forms of DPHL rather than examples of a different disease. They meet the basic requirements for the diagnosis of the syndrome, i.e., they were delayed, post-hypoxic and leukoencephalopathic, as determined by the dSIR sequences.

### 5.14. Conclusions

As a result of the use of dSIR pulse sequences, it is possible that DPHL will become a commonly recognized complication of severe drug overdoses and other causes of asphyxia rather than be regarded as a rare syndrome. However, further studies will be needed to establish this.

The opioid epidemic with a less-than-ideal initial treatment of patients’ overdoses is likely to be producing an increase in the incidents of cases of DPHL. This puts a premium on the accurate recognition and diagnosis of the syndrome to assist in patient management.

## Figures and Tables

**Figure 1 diagnostics-14-00418-f001:**
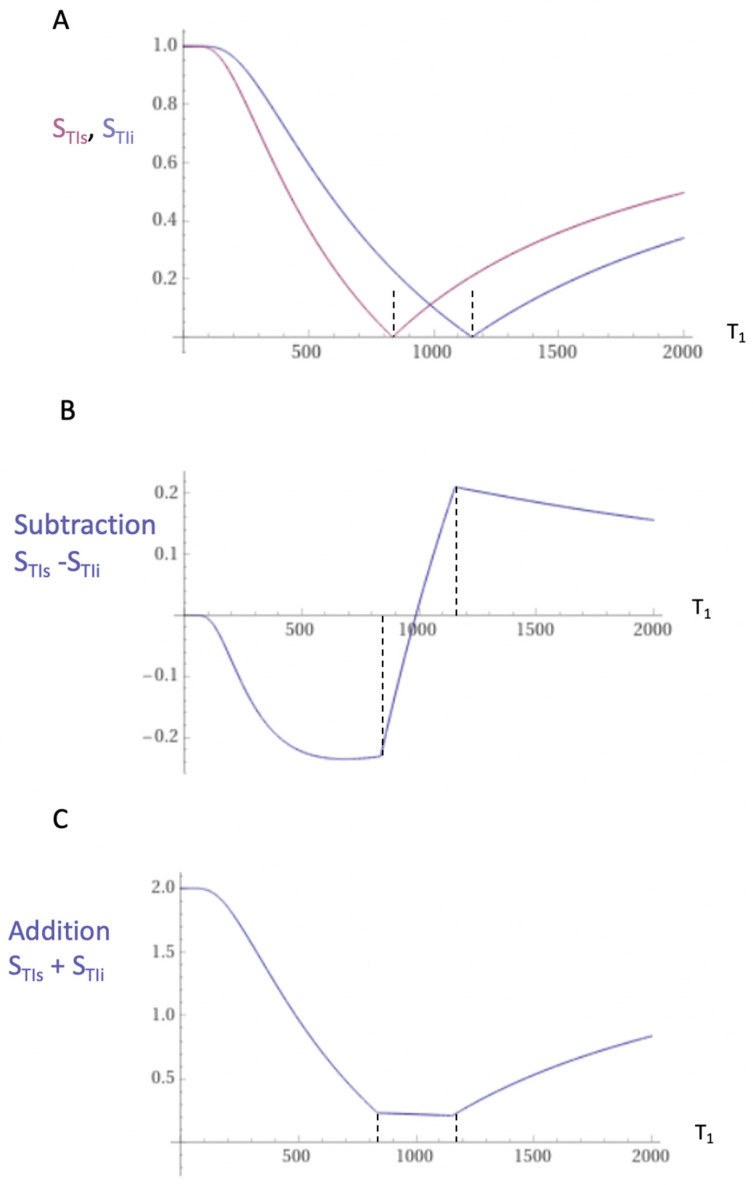
Subtracted IR (SIR) and Added IR (AIR) T_1_ filters. T_1_ is shown along the linear X axes in ms. (**A**) shows the TI_s_ T_1_ filter (pink) and TI_i_ T_1_ filter (blue), (**B**) shows the subtraction (S_TIs_ − S_TIi_) IR or SIR T_1_ filter and (**C**) shows the addition (S_TIs_ + S_TIi_) IR or AIR T_1_ filter. The middle Domain (mD) is the T_1_ values along the X axis between the vertical dashed lines. In (**B**), the slope of the curve in the mD in the SIR T_1_ filter is about double that of the S_TIs_ filter (pink in [A]). In (**C**), the signal at T_1_ = 0 is doubled to 2.0, and the signal in the mD is reduced to about 0.20 in the nearly linear, slightly downward sloping central part of the AIR T_1_ filter (i.e., in the middle Domain, mD).

**Figure 2 diagnostics-14-00418-f002:**
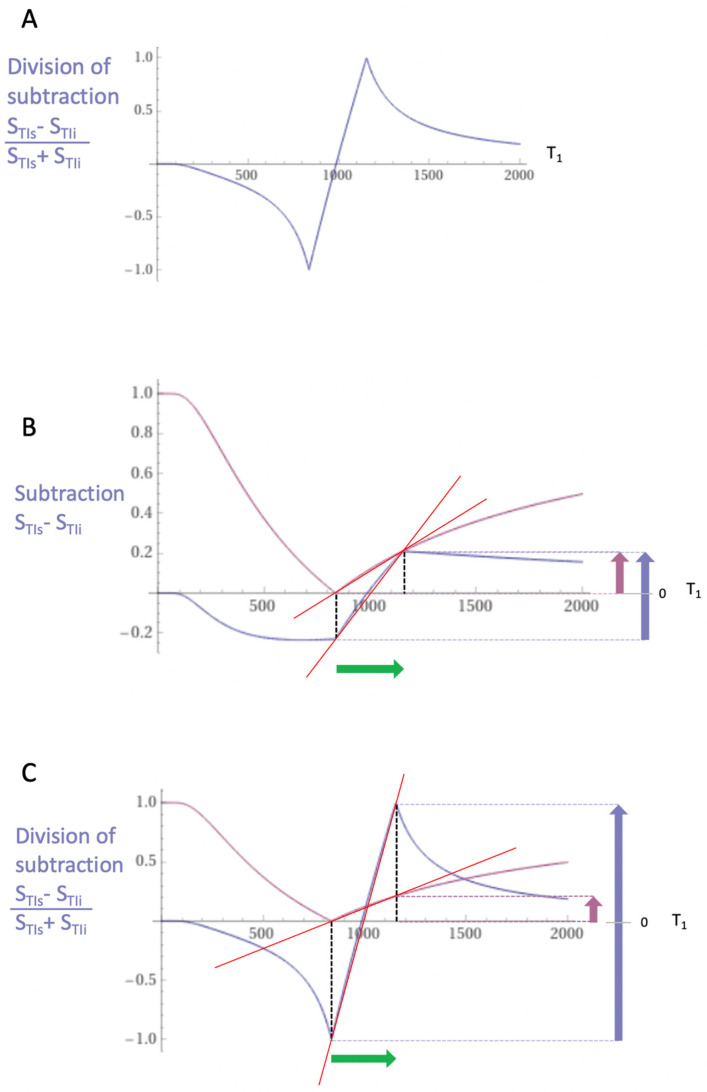
(**A**) shows a division of the SIR T_1_-filter in Figure 1B by the additional T_1_ filter in Figure 1C to give the dSIR T_1_-bipolar filter. (**B**) shows a comparison of the conventional IR S_TIs_ T_1_ filter (pink) and the SIR T_1_ filter (blue) for a small increase in T_1_ (horizontal green arrow, ∆T_1_). (**C**) is a comparison of the S_TIs_ T_1_ filter (pink) with the dSIR T_1_ filter (blue) for the small increase in T_1_. Multiplication of the small increase in T_1_ (∆T_1_) by the slope of the filters (red lines) produces the contrast shown as vertical arrows on the right (bounded by the horizontal dotted lines). In (**B**), the contrast produced by the SIR T_1_ filter is twice that produced by the IR T_1_ filter (blue and pink arrows). In (**C**), the contrast produced by the dSIR T_1_-bipolar filter is ten times greater than that produced by the IR T_1_-filter (blue and pink arrows).

**Figure 3 diagnostics-14-00418-f003:**
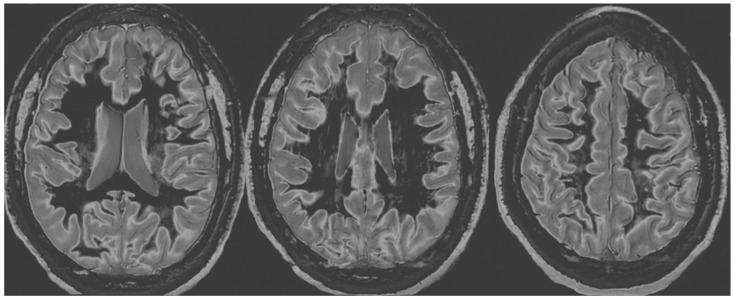
Normal volunteer. Two-dimensional narrow mD dSIR images. The narrow mD dSIR images show normal white matter as a very low signal intensity (dark) except for intermediate areas in and around the corticospinal tracts. Normal high signal boundaries are seen at the junction between white matter and grey matter.

**Figure 4 diagnostics-14-00418-f004:**
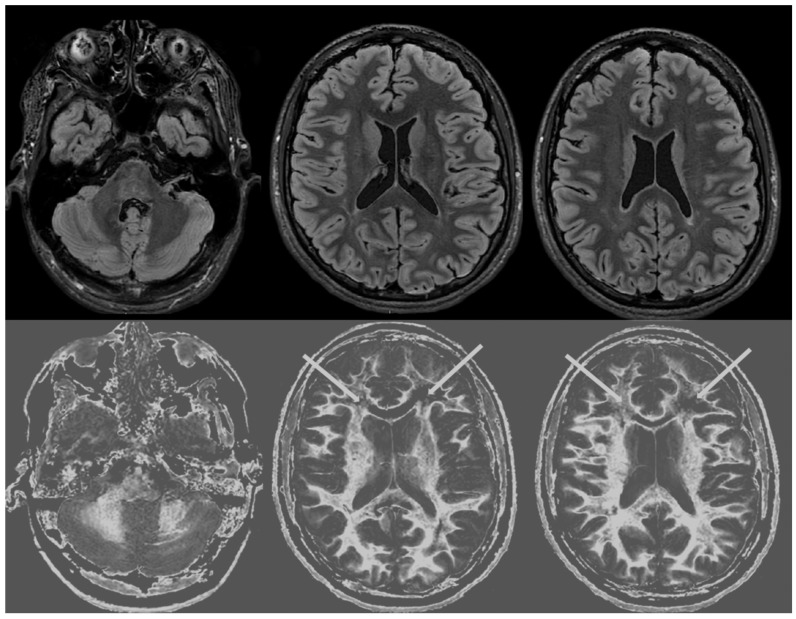
Case 1 was examined nine months after a severe drug overdose. Comparison of positionally matched T_2_-FLAIR images (upper row) and narrow mD dSIR images (lower row). No abnormality is seen on the T_2_-FLAIR image, but there are extensive areas of a high signal in the white matter of the brain. Only the anterior and posterior central corpus callosum and parts of the frontal lobes have a low signal (dark appearance) and look normal on the dSIR images (white arrows) (lower row). High signal boundaries are seen at the junction between the white matter and grey matter but are rendered less obvious in many areas because of the high signal in much of the white matter.

**Figure 5 diagnostics-14-00418-f005:**
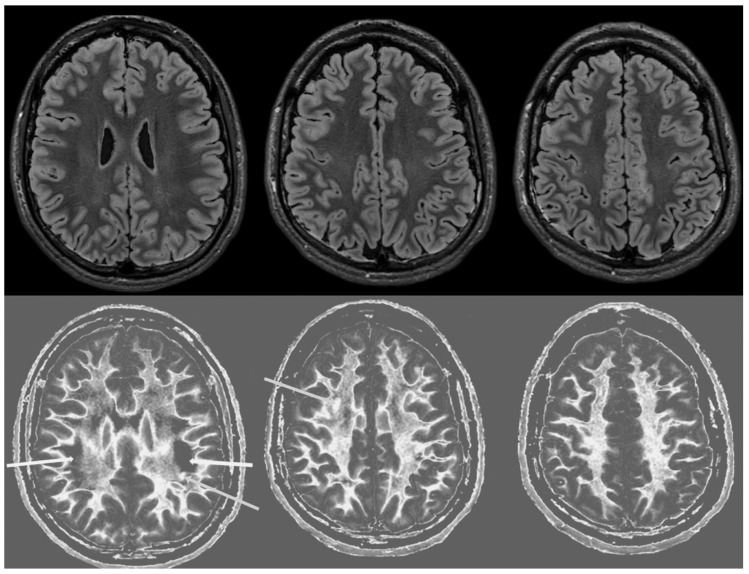
Case 1 was examined nine months after a severe drug overdose. Comparison of positionally matched T_2_-FLAIR images (upper row) and narrow mD dSIR images (lower row) (higher cerebral hemispheres). No abnormality is seen on the T_2_-FLAIR images, but there are extensive areas of high signals in the central white matter of the brain (lower row). Only some of the peripheral white matter on the lower images appears dark and looks normal on the dSIR images (white arrows) (lower row). Some other areas of white matter have a mid-grey appearance consistent with a lesser degree of abnormality. Small focal lesions are also seen on the dSIR images (grey arrows) but not on the T_2_-FLAIR images.

**Figure 6 diagnostics-14-00418-f006:**
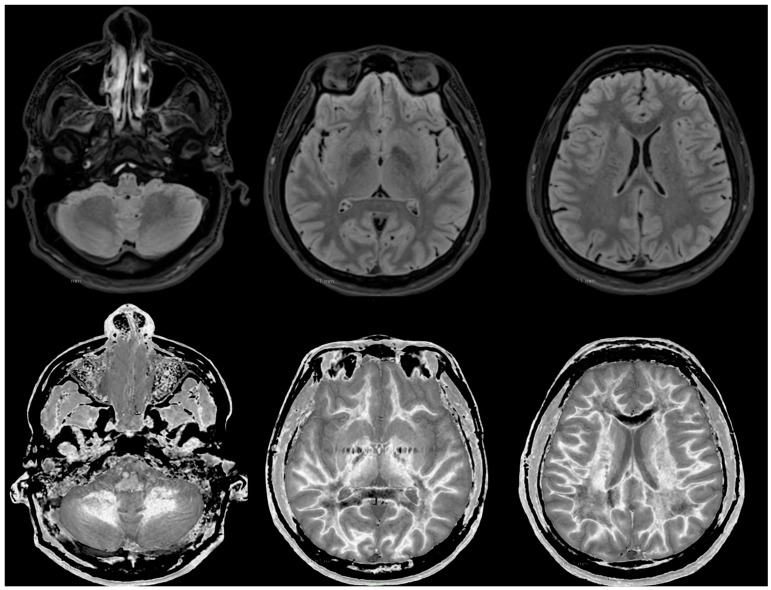
Case 2 was examined two years after his asphyxial episode. Comparison of positionally matched T_2_-FLAIR images (upper row) and narrow mD dSIR images (lower row). No abnormality is seen on the T_2_-FLAIR images, but there are extensive areas of high signals in the white matter of the brain. The anterior and posterior central corpus callosum and parts of the frontal lobes have a lower, more normal signal (darker appearance). Horizontal CSF flow artefacts are seen in the dSIR image in the middle lower row.

**Figure 7 diagnostics-14-00418-f007:**
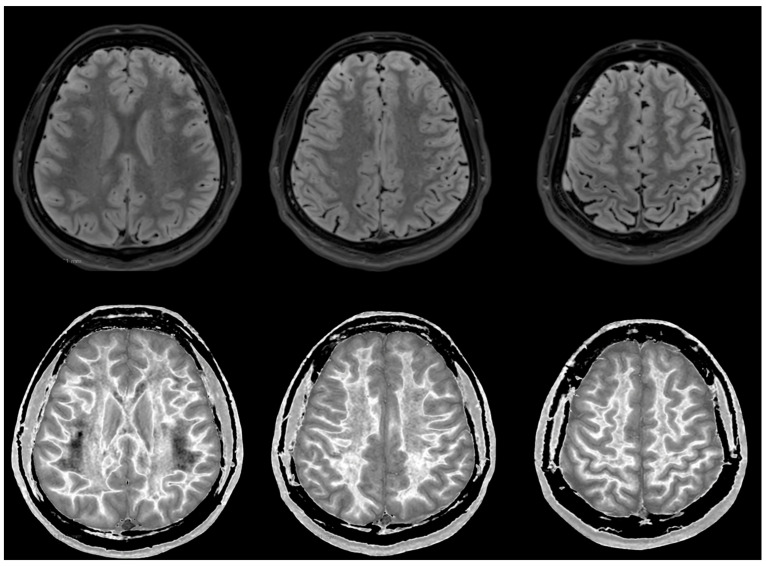
Case 2 was examined two years after his asphyxial episode. Comparison of positionally matched T_2_-FLAIR images (upper row) and narrow mD dSIR images (lower row). No abnormality is seen on the T_2_-FLAIR images, but there are extensive areas of high signals in the central white matter of the brain (lower row). Only some of the peripheral white matter on the lower images appears darker and looks more normal on the dSIR images.

**Table 1 diagnostics-14-00418-t001:** Pulse sequences and pulse sequence parameters used at 3T. Z = zipped.

#	Sequence	TR (ms)	TI (ms)	TE (ms)	Matrix Size Voxel Sizes (mm)	Number of Slices	Slice Thickness (mm)
1	2D FSE IR (for white matter nulling)	9192	350	7	256 × 224 0.9 × 0.1 Z512 0.4 × 0.4	26	4
2	2D FSE IR (used with #1 for narrow mD dSIR)	5796	500	7	256 × 224 0.9 × 0.1 Z512 0.4 × 0.4	26	4
3	2D and 3D T_2_-FLAIR	6300	1851	102	320 × 240 240 × 240 0.7 × 0.7, 1 × 1 Z512 0.5 × 0.5	26, 122	4, 2

## Data Availability

The data presented in this study are available on request from the corresponding author due to privacy restrictions.

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
