# Peer review of "Diagnosis of Delayed Post-Hypoxic Leukoencephalopathy (Grinker’s Myelinopathy) with MRI Using Divided Subtracted Inversion Recovery (dSIR) Sequences: Time for Reappraisal of the Syndrome?"

_diagnostics, 2024, doi:10.3390/diagnostics14040418_

Round 1

Reviewer 1 Report

Comments and Suggestions for Authors

1. It would be helpful to provide more background information on the condition of delayed post-hypoxic leukoencephalopathy, including its aetiology, prevalence, and symptoms. 

2. The paper could benefit from a more in-depth discussion of the limitations and challenges associated with diagnosing this condition, particularly with MRI techniques. 

3. It would be useful to include a comparison of dSIR sequences with other MRI techniques commonly used for diagnosing post-hypoxic leukoencephalopathy, such as FLAIR or DWI. 

4. In addition to the images provided, it would be beneficial to include a description and interpretation of the specific features observed on dSIR sequences in patients with this condition. 

5. The study sample size may be limited, and it would be helpful to acknowledge this and discuss potential implications for the generalizability of the findings. 

6. The conclusions drawn from the results of the study could be strengthened by including statistical analyses and providing confidence intervals for the observed measurements. 

7. There should be a discussion on the potential benefits of using dSIR sequences over traditional MRI techniques for diagnosing delayed post-hypoxic leukoencephalopathy, including considerations for patient comfort and scan time. 

8. It would be informative to include a brief overview of treatment options for this condition and how early diagnosis through dSIR sequences may impact patient outcomes. 

9. The paper could benefit from a clearer explanation of the methodology used for obtaining and interpreting the dSIR images, including any potential limitations or sources of error. 

10. It would be beneficial to include case studies or examples of patients with confirmed delayed post-hypoxic leukoencephalopathy and their corresponding dSIR images to demonstrate the practical application of the findings. 

Author Response

Response to Referee 1's comments and suggestions:

  1. Thank you for the suggestion to provide more background information of delayed post-hypoxic leukoencephalopathy including its aetiology, prevalence and symptoms. ------We have mentioned that it is rare, cited the MR imaging studies from 1991-2015 where 24/41 patients had CO poisoning and 15/41 had drug overdoses. These are the commonest causes. A very small number were due to asphyxia (as with Case 2 in the paper). We also mentioned the classic symptoms i.e. (i) Parkinsonism and (ii) Akinetic mutism as well as other symptoms that may be seen.

  1. "The paper could benefit from a more in-depth discussion of the limitations and challenges associated with diagnosing this condition, particularly with MRI techniques."

We mentioned the outstanding limitation of conventional MR techniques i.e. the techniques such as T2-FLAIR are negative when dSIR sequences are positive.

  1. "It would be useful to include a comparison of dSIR sequences with other MRI techniques commonly used for diagnosing post-hypoxic leukoencephalopathy, such as FLAIR or DWI."

We included a comparison with T2-FLAIR (Figs. 4-7). We also mentioned that DWI has only been used inconsistently in previous studies of classical DPHL and that it showed restriction in some cases but not all. Future more extensive comparisions with DWI may be helpful.

  1. "In addition to the images provided, it would be beneficial to include a description and interpretation of the specific features observed on dSIR sequences in patients with this condition."

We have described the specific changes in the text and legends as well as under the general description of the whiteout sign. We have also mentioned that we believe these changes are due to widespread small increases in T1 in abnormal white matter.

  1. "The study sample size may be limited, and it would be helpful to acknowledge this and discuss potential implications for the generalizability of the findings."

We mention that DPHL is a rare condition, and that the literature largely consists of single case reports or series of 2-5 cases, so that in this context the two cases reported in this paper is typical for the condition. We have included an extra sentence in the conclusion: "However, further studies will be needed to establish this."

  1. "The conclusions drawn from the results of the study could be strengthened by including statistical analyses and providing confidence intervals for the observed measurements."

There is probably limited scope for statistics in two cases when the result in each was negative with T2-FLAIR and positive with dSIR. There were no observed measurements to provide confidence intervals on.

  1. "There should be a discussion on the potential benefits of using dSIR sequences over traditional MRI techniques for diagnosing delayed post-hypoxic leukoencephalopathy including considerations for patient comfort and scan time."

We have mentioned that the main benefit of the dSIR sequence is that we see abnormalities that are not seen with traditional sequences. The sequences take less than 5 minutes so the time is similar to conventional sequences. The impact on patient comfort and scan time is therefore thought to be minimal.

  1. "It would be informative to include a brief overview of treatment options for this condition and how early diagnosis through dSIR sequences may impact outcomes.

We have stated that the main benefit may be diagnosis of unrecognized brain disease as a consequence of hypoxia in patients with psychiatric diagnoses, and point out in Case 2 that there was no medical recognition of his underlying brain disease for about two years. Treatment options and limitations may follow from the recognition of the presence of organic brain disease in the patients.

  1. "The paper could benefit from a clearer explanation of the methodology used for obtaining and interpreting the dSIR images, including any potential limitations or sources of error."

We provided two references (15, 16) describing the sequence in detail, a system for radiological interpretation and failure modes for the sequence. The latter includes limitations and sources of error in detail. The two papers are quite lengthy (33 and 34 pages respectively) and including the level of detail in these papers would distort the submitted paper, and be repetitive.

  1. "It would be beneficial to include case studies or examples of patients with confirmed delayed post-hypoxic leukoencephalopathy and their corresponding dSIR images to demonstrate the practical application of the findings."

We have included two cases of presumptive DPHL and explained why we believe that these are in fact mild cases of classic DPHL even though they do not meet the classical criteria for diagnosis of DPHL (i.e. Parkinsonism/akinetic mutism, and abnormalities on T2-wSE and/or T2-FLAIR images). Because classical DPHL is a rare syndrome we might have to wait some years to find even a single case of classic DPHL to image with dSIR sequences to provide data on a confirmed case of DPHL (using classical criteria). A series might take longer.

Reviewer 2 Report

Comments and Suggestions for Authors

This is an in innovative study that is adequately presented both in terms of theory and methodology. The two cases of DPHL that have been presented seem to provide strong credibility to your conclusions on the potential expanded role of dSIR technique in clinical MRI practice.

Comments on the Quality of English Language

The manuscript is very well-written. I suggest several minor linguistic changes as follows:

In Discussion, p11, l266 - "respectively" is superfluous, I suggest to delete it.

In Discussion, p12, l324 - "demyelination" instead of "myelination".

In Discussion, p12, l325 - "astrocyte" instead of "astrolyte"

In Discussion, p12, l326 - I presume you mistyped "oligodendroglia" as "oligodendroma"?

Author Response

Response to Referee 2's comments and suggestions:

  1. "This is an innovative study that is adequately presented both in terms of theory and methodology. The two cases of DPHL that have been presented seem to provide strong credibility to your conclusions on the potential role of dSIR technique in clinical MRI practice."

We thank the referee for his or her assessment.

  1. "The manuscript is very well-written. I suggest several minor linguistic changes as follows:"

Thank you for well-spotted errors. We have corrected them.

–––––––––––––––––––––––––––––––––––––

Please note the following suggestions from our previous email:

Page 6, line 143-147: Please make Figures 3-7 of the same width and align them in the same way within the margins of the pages. If possible, maximize the size of the images. Figure 3 makes full use of the margins and has the images as large as practical. This could serve as a template for images 4-7.

Page 7, line 182: as above

Page 8, line 190: as above

Page 10, line 250: as above

Page 11, line 257: as above.

–––––––––––––––––––––––––––––––––––––

Fonts and type sizes

In addition, the fonts and type sizes are inconsistent throughout the proofs e.g. Palatino linotype and Times New Roman fonts are mixed, even within the same section. Headings and text are not of a consistent font size.

Round 2

Reviewer 1 Report

Comments and Suggestions for Authors

All the mentioned comments were addressed. The present form has eligible for publication.